# INTERACTION-AWARE 4D GAUSSIAN SPLATTING FOR DYNAMIC HAND-OBJECT INTERACTION RECONSTRUCTION

## ABSTRACT

This paper focuses on a challenging setting of simultaneously modeling geometry and appearance of hand-object interaction scenes without any object priors. We follow the trend of dynamic 3D Gaussian Splatting based methods, and address several significant challenges. To model complex hand-object interaction with mutual occlusion and edge blur, we present interaction-aware hand-object Gaussians with newly introduced optimizable parameters aiming to adopt piecewise linear hypothesis for clearer structural representation. Moreover, considering the complementarity and tightness of hand shape and object shape during interaction dynamics, we incorporate hand information into object deformation field, constructing interaction-aware dynamic fields to model flexible motions. To further address difficulties in the optimization process, we propose a progressive strategy that handles dynamic regions and static background step by step. Correspondingly, explicit 3D regularizations are designed to stabilize the hand-object representations for smooth motion transition, physical interaction reality, and coherent lighting. Experiments show that our approach surpasses existing dynamic 3D-GS-based methods and achieves state-of-the-art performance in reconstructing dynamic hand-object interaction.

## 1 INTRODUCTION

Accurate hand–object interaction (HOI) reconstruction is vital for VR and robotics Handa et al. (2020), requiring precise shape modeling and interaction capture. Despite the apparent simplicity of daily actions like grasping or drinking, they involve complex contact dynamics and severe occlusions that remain challenging to model.

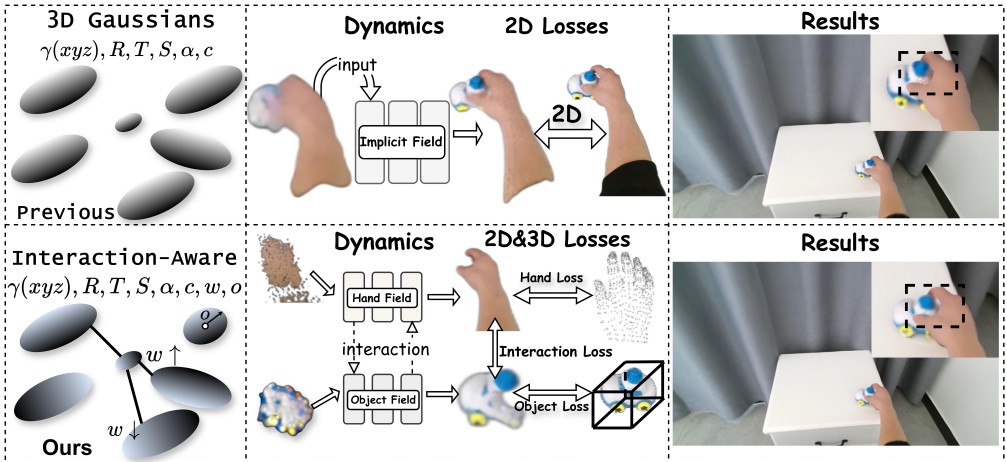

Fig. 1. **Differences between traditional 3D Gaussian-based hand-object reconstruction and our interaction-aware modeling.** Prior methods predominantly rely on 2D supervision and employ a single implicit field for dynamic estimation, frequently failing to capture fine-grained interaction details. (e.g., the exact clearance between the toy car and hand). In contrast, our interaction-aware approach simultaneously constructs the hand field and interaction-aware object field, applying 2D&3D losses to accurately model the details of hand-object interaction.

Previous works Rong et al. (2020); Taheri et al. (2020) reconstruct HOI scenes by treating interactive objects as known, relying on specific object poses or templates. However, acquiring such poses or templates is costly, limiting their industrial applicability. Existing methods try to reduce the reliance on precise pose estimation and templates by several routes. Qi et al. (2024); Wen et al. (2023) employs SDF-based approaches, integrating SDF to reconstruct dynamic hand-object scenes using neural networks, without the need for specific object priors. However, these methods focus primarily on geometry reconstruction without appearance. With the advent of NeRF Mildenhall et al. (2021), some researchers Liu et al. (2023); Zhang et al. (2023) explore HOI scene reconstruction with both geometry and appearance by training implicit fields. However, due to the inefficiency of backward mapping-based ray-rendering algorithms Mildenhall et al. (2021), these methods require significant time and computational resources. Recently, 3D Gaussian Splatting (3D-GS) Kerbl et al. (2023) has demonstrated superior fidelity and speed in static scene reconstruction. Some works Wu et al. (2024); Huang et al. (2024); Yang et al. (2024) attempt dynamic reconstruction using 3D-GS, but they struggle with complex HOI scenarios involving heavy occlusion and irregular rotations, failing to capture accurate interaction dynamics. Although EgoGaussian Zhang et al. (2024) targets HOI reconstruction, it requires object pose estimation and only presents results for interactive objects without effective hand representation. In this work, we address the limitations of 3D-GS-based methods by proposing a model to simultaneously reconstruct the entire HOI scene without requiring any object priors.

To successfully handle such a practical setting presents significant challenges. First, drastic motions, mutual occlusion and blur during interaction cause misalignment and excessive overlap among Gaussians. To address this, we model the interaction as a piecewise linear process and present a novel representation termed interaction-aware hand-object Gaussians. It introduces two parameters over the traditional 3D-GS representation: weight $w$ and radius $o$. The weight $w$ balances motion smoothness and noise reduction, with smaller values indicating weak structural information or occlusion. The radius $o$ controls edge sharpness, where smaller values produce clearer contours. The combination of $w$ and $o$ effectively models the complex dynamic interaction, reducing blurring at interaction boundaries and enhancing visual quality. Second, previous methods Huang et al. (2024); Yang et al. (2024) use a single field to model Gaussian transformations, which is insufficient for capturing drastic motions in HOI scenes, leading to loss of motion details. On the other hand, simply using separate fields for hand and object deformation overlooks their mutual interaction. To address this, we incorporate key-frame hand positions into the object field, enabling interaction-aware transformations that capture dynamic changes caused by hand grasping. Third, considering flexible motions and irregular rotations in HOI scenes, it is difficult to directly utilize traditional 3DGS optimization Kerbl et al. (2023); Wu et al. (2024); Yang et al. (2024) to achieve decent rendering quality. To address this, we design 3D regularizations to explicitly stabilize the position and rotation of hand-object Gaussians. Furthermore, we propose a progressive optimization mechanism to achieve physically reality of hand-object interaction, smooth edge transitions and enhance lighting coherence in HOI scenes.

Our contributions are summarized as follows:

**1.** We propose a novel interaction-aware hand-object Gaussian representation to model HOI scenes without any object priors, effectively addressing mutual occlusion and edge blur during interactions.
**2.** To accurately model interaction changes on the hand-held object, we incorporate hand information to enhance the object field to represent relevant deformation.
**3.** We employ a progressive optimization strategy with explicit 3D losses to benefit the fitting of the interaction-aware Gaussians during dynamic reconstruction.
**4.** Experiments show that our approach surpasses existing dynamic 3D-GS-based methods Yang et al. (2024); Wu et al. (2024); Huang et al. (2024), achieving state-of-the-art performance in reconstructing dynamic HOI scenes.

## 2 RELATED WORKS

**Hand Representation.** Early approaches Mueller et al. (2018; 2019) focused on estimating 2D or 3D keypoints from images. The introduction of statistical hand models like MANO Romero et al. (2022) revolutionized parametric hand representation by jointly encoding pose, shape, and 3D vertices. Recent methods Baek et al. (2019) typically employ regression networks to predict MANO parameters directly from images and optimize shape parameters for alignment. However,

these methods suffer from error propagation: initial MANO inaccuracies accumulate downstream, causing cascading reconstruction errors. To mitigate this, we propose a lightweight hand field to decouple hand representation from strict MANO parameter dependencies.

**Hand-Object Reconstruction.** Reconstructing hand-object interaction from video remains a significant challenge in computer vision and graphics. Previous works fall into two categories. The first Hampali et al. (2020); Moon et al. (2020) reconstructs hand-object interactions from multiview sources by fitting objects into 2D images using 3D object priors. However, these methods heavily rely on accurate 3D priors, which are costly to obtain. The second stream Rong et al. (2020); Fan et al. (2023); Hasson et al. (2019) pre-learns object templates to reduce reliance on priors. For example, the MANO model Romero et al. (2022) represents the canonical hand space, with linear blending skinning driving the hand template. EgoGaussian Zhang et al. (2024) reconstructs egocentric interaction scenes using 3D Gaussian Splatting by separating dynamic objects from the static background. However, it is sensitive to object pose, probably failing under inaccurate poses, and excludes interacting hands from reconstruction. As a result, it misses the complete hand–object interaction context. Recent work BIGS On et al. (2025) uses 3DGS and a pre-trained diffusion model for bimanual HOI from monocular video but assumes known object mesh and focuses on two-handed interactions, our method is category-agnostic and requires no object priors.

**Dynamic Scene Reconstruction.** With the advent of NeRF Mildenhall et al. (2021), many works Pumarola et al. (2021); Guo et al. (2023); Li et al. (2021); Park et al. (2021) use MLPs to represent implicit spaces as deformation fields with temporal information. However, their extensive training time limits practical applicability. 3D Gaussian Splatting (3D-GS) Kerbl et al. (2023) emerges as a promising alternative for scene reconstruction. Methods like Wu et al. (2024); Yang et al. (2024); Huang et al. (2024) explore dynamic reconstruction using 3D-GS. For instance, Yang et al. (2024) uses MLPs to learn Gaussian position offsets per timestamp, which increases training time. Huang et al. (2024) introduces sparse control points to deform Gaussians, but in hand-object interaction (HOI) scenes, redundant points lead to inaccuracies and image tearing, failing to capture intricate interactions. These methods Wu et al. (2024); Yang et al. (2024); Huang et al. (2024) input all Gaussians into a single MLP, which struggles to accurately model complex interactive motions. To overcome significant challenges posed by HOI scenes, our method introduces a novel interaction-aware hand-object Gaussian representation, with adaptive losses and a progressive optimization strategy.

## 3 METHOD

Our goal is to reconstruct dynamic hand-object interaction (HOI) scenes from RGB egocentric videos at *arbitrary timestamps* without relying on any object *shape priors*. We utilize three implicit fields to model the dynamic HOI scenes: the hand field $\mathcal{F}_H$ and object field $\mathcal{F}_{object}$ to approximate the shape of the dynamic HOI region, as well as the background field $\mathcal{F}_\theta$ to create a clean background and facilitate subsequent joint optimization. Separate modeling allows capturing clear hand-object appearance and stable background scene in drastically changing dynamic scenarios. First, by treating hand and object modeling differently, significant occlusions could be solved via more detailed supervision. Second, backgrounds require low-frequency updates, while hand-object interactions demand high-frequency modeling. Meanwhile, collaborating with such dynamic implicit fields, we adaptively improve the Gaussian Splatting representation for HOI scenarios, addressing occlusion and contour clarity issues during the interaction. In optimization, we utilize explicit 3D information provided by MANO parameters Romero et al. (2022) and the 3D object bounding boxes, facilitating significantly faster and more stable convergence. Moreover, we employ an interaction loss to ensure the physical reality of the interaction. A progressive and collaborative optimization framework is devised to achieve high quality HOI scene reconstruction with such 3D explicit supervision and elaborated interaction-aware representation.

### 3.1 PRELIMINARIES: DEFORMABLE GAUSSIAN SPLATTING

3D Gaussian Splatting (3D-GS) Kerbl et al. (2023) represents 3D scene features using five parameters: position, transparency, spherical harmonic coefficients, rotation, and scaling. 3D-GS explicitly defines each 3D Gaussian ellipsoid in space using a covariance matrix $\Sigma$ and the position vector $\rho$,

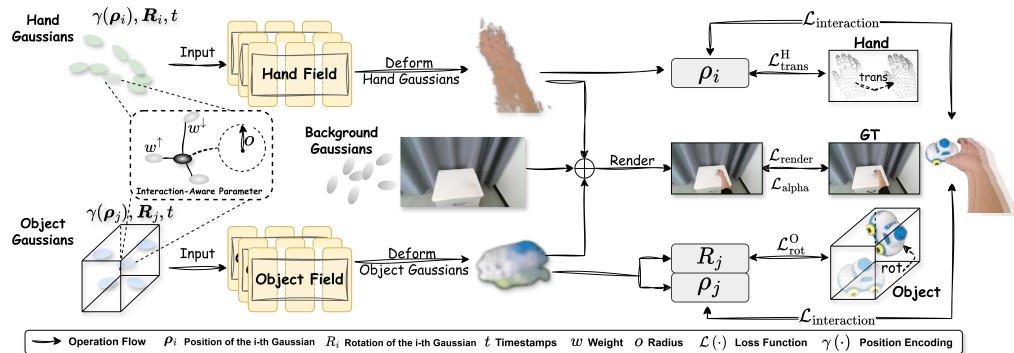

Fig. 2. **Overview of interaction-aware hand-object Gaussians.** We propose a novel framework for reconstructing dynamic HOI scenes from RGB videos without object shape priors. The framework consists of three components: (1) *Specialized Implicit Fields*: separate hand, object, and background fields disentangle dynamic interactions, with hand/object fields capturing high-frequency deformations and occlusions (leveraging hand information for object's interaction-aware deformation) while the background field maintains low-frequency stability; (2) *Interaction-aware Gaussian Splatting*: enhances representation with adaptive weights $w$ and radius $o$ to address contour ambiguity and occlusions; (3) *Progressive Optimization*: combines explicit 3D supervision with physical interaction constraints for efficient convergence.

as shown in the following equation:

$$\mathcal{G}(\boldsymbol{x}) = \frac{1}{(2\pi)^{\frac{3}{2}} |\Sigma|^{\frac{1}{2}}} \exp(-\frac{1}{2}(\boldsymbol{x}-\boldsymbol{\rho})^\top \Sigma^{-1}(\boldsymbol{x}-\boldsymbol{\rho})), \tag{1}$$

here the $\Sigma$ matrix can be decomposed into a rotation $\boldsymbol{R}$ and a scaling $\boldsymbol{S}$ by $\Sigma = \boldsymbol{R}\boldsymbol{S}\boldsymbol{S}^\top\boldsymbol{R}^\top$.

Recent works Yang et al. (2024); Wu et al. (2024); Huang et al. (2024) combine 3D-GS with deformation fields for dynamic scenes, using an MLP to warp points from canonical to target space:

$$\begin{aligned} \mathcal{F}_\theta(\boldsymbol{x}, \boldsymbol{t}) &= (\delta\boldsymbol{x}^t, \delta\boldsymbol{r}^t, \delta\boldsymbol{s}^t), \\ \boldsymbol{x}^t &= \boldsymbol{x} + \delta\boldsymbol{x}^t. \end{aligned} \tag{2}$$

Each SfM-initialized 3D Gaussian's center $\boldsymbol{x}$ is input to deformation field $\mathcal{F}_\theta$, which outputs time-dependent offsets $(\delta\boldsymbol{x}^t, \delta\boldsymbol{r}^t, \delta\boldsymbol{s}^t)$ to wrap canonical Gaussians to target space.

## 3.2 INTERACTION-AWARE HAND-OBJECT GAUSSIANS

To effectively capture the complex spatiotemporal motion in the Hand-Object Interaction (HOI) scene, we propose to decompose the dynamic HOI scene into three sets of Gaussians and model each part individually. Moreover, we improve traditional 4D Gaussian representations to overcome the issues representing complex HOI motions in dynamic scenarios. Traditional 4D Gaussians Wu et al. (2024); Yang et al. (2024); Huang et al. (2024) tend to neglect mutual influences between different interacted Gaussians. Moreover, it is hard to depict contour edges in interaction process, leading to texture drift and edge blur. Inspired by Huang et al. (2024), we introduce two additional learnable parameters weight $\boldsymbol{w} \in \mathbb{R}^+$ and radius $\boldsymbol{o} \in \mathbb{R}^+$, forming a novel representation termed the Interaction-Aware Hand-Object Gaussian $\mathcal{G}_{\text{HO}}$. $\mathcal{G}_{\text{HO}}$ focuses on interaction-aware modeling via: (1) Weight $\boldsymbol{w}$ smooths the motion and reduces noise, it models mutual occlusion during interactions, where a small weight $\boldsymbol{w}$ indicates weak structural information and occlusion; (2) Radius $\boldsymbol{o}$ captures edge details, where a small radius $\boldsymbol{o}$ corresponds to sharper geometric contours near edges; (3) Since $\boldsymbol{w}$ is larger near the current Gaussian and smaller farther from the edge, the combination of weight $\boldsymbol{w}$ and radius $\boldsymbol{o}$ effectively handles edge blurring between hand-object interactions and the background. $\mathcal{G}_{\text{HO}}$ is expressed as follows:

$$\mathcal{G}_{\text{HO}} = \{\boldsymbol{x}_i\boldsymbol{y}_i\boldsymbol{z}_i, \boldsymbol{R}_i, \boldsymbol{T}_i, \boldsymbol{S}_i, \boldsymbol{\alpha}_i, \boldsymbol{c}_i, \boldsymbol{w}_i, \boldsymbol{o}_i\}. \tag{3}$$

Due to different characteristics of hand motions, object motions and background scenes, we separately introduce modeling of the dynamics of each component below.

**Hand Gaussians.** Hand Gaussians $\mathcal{G}_{\text{H}}$ has the same optimizable parameters with $\mathcal{G}_{\text{HO}}$, and is modeled with a hand-implicit field $\mathcal{F}_{\text{H}}$ to capture the time-varying transformation of hand motion. This

field $\mathcal{F}_{\mathrm{H}}$ takes the timestamp $t$ and the canonical position $(\boldsymbol{x}_i \boldsymbol{y}_i \boldsymbol{z}_i)$ of the $i$-th hand Gaussian as inputs. Since our setting requires modeling dynamic HOI scenes at any time from any view pose, we use $t$ as the input and construct the following formula (details provided in the appendix):

$$\Delta \mathcal{G}_{\mathrm{H}} = \mathcal{F}_{\mathrm{H}} \left\{ \gamma \left( \boldsymbol{x}_i \boldsymbol{y}_i \boldsymbol{z}_i \right), \gamma \left( t \right) \right\}, \tag{4}$$

$\gamma(\cdot)$ is positional encoding Bhatnagar et al. (2020); Yang et al. (2024). Adding noise$_{\mathrm{smooth}}$ Yang et al. (2024) to $\gamma(t)$ prevents oversmoothing and retains hand details while fitting coarse geometry.

**Object Gaussians.** Hand-object interactions often cause deformations (e.g., squeezing) or occlusions (e.g., holding). To enhance the ability of the object field $\mathcal{F}_{\mathrm{O}}$ to capture interaction-aware deformations, we introduce hand position as an additional input to the object field. This overcomes the limitation of implicit methods Wu et al. (2024); Yang et al. (2024), which generate global offsets without explicit hand-object interaction modeling. The object field $\mathcal{F}_{\mathrm{O}}$ takes both hand and object positions as inputs, formulated as follows (details in appendix):

$$\Delta \mathcal{G}_{\mathrm{O}} = \mathcal{F}_{\mathrm{O}} \left\{ \gamma \left( \left( \boldsymbol{x}_i^k \boldsymbol{y}_i^k \boldsymbol{z}_i^k \right) \oplus \left( \boldsymbol{x}_j^k \boldsymbol{y}_j^k \boldsymbol{z}_j^k \right) \right), \gamma \left( t \right) \right\}. \tag{5}$$

Here, $\oplus$ concatenates hand $(\boldsymbol{x}_i^k, \boldsymbol{y}_i^k, \boldsymbol{z}_i^k)$ and object $(\boldsymbol{x}_j^k, \boldsymbol{y}_j^k, \boldsymbol{z}_j^k)$ positions with the canonical Gaussian position at key-frame $k$—the moment just before hand-object interaction. The object-implicit field $\mathcal{F}_{\mathrm{O}}$ predicts time-varying offsets $\Delta \mathcal{G}_{\mathrm{O}}$ at timestamp $t$, and uses linear annealing of noise$_{\mathrm{smooth}}$ Yang et al. (2024) to stabilize training.

**Background Gaussians.** We construct background Gaussians to better capture the smoothness and static nature of the background and avoid the unstable dynamic changes of the background Gaussian distribution caused by the interaction of foreground hand-object, which will affect the rendering quality of the background Huang et al. (2024); Wu et al. (2024). Background Gaussians $\mathcal{G}_{\mathrm{BG}}$ are based on the Deformable-3DGS model Yang et al. (2024). Their positions change over time, as formulated in Eq. equation 2, using the background-implicit field $\mathcal{F}_{\mathrm{BG}}$ with timestamps $t$.

### 3.3 EXPLICIT 3D-2D REGULARIZATIONS

2D regularization is to constrain pixel errors in image space. However, this is insufficient due to significant occlusion and drastic motion for hand-object interaction scenes. To enable Gaussians to accurately and efficiently model complex hand-object interactions, besides 2D supervision, we introduce explicit 3D regularizations. These include object, hand and interaction loss to stabilize the rotation and transformation of interaction-aware hand-object Gaussians.

**Object Loss.** By feeding hand position into the object field, object translation aligns naturally with hand motion. Translation is easily constrained via hand pose, but rotation remains difficult. Passive object rotation in interactions often causes distortion or flipping, making rotation constraints essential. We introduce a rotation consistency constraint to ensure that the motion of interaction Gaussians locally adheres to physically plausible rigid transformations. Let $\mathbf{R}_0 \in \mathrm{SO}(3)^N$ denote the rotation of $N$ Gaussians at the initial time $t_0$, and $\mathbf{R}_t \in \mathrm{SO}(3)^N$ denote their predicted rotations at a target time $t$. Concurrently, from the positional trajectories of the Gaussians, we compute a geometry-induced rigid rotation field $\mathbf{R}_{\mathrm{arap}} \in \mathrm{SO}(3)^N$ via local neighborhood rigid alignment (solved in closed form using SVD), defined as the solution to:

$$\mathbf{R}_{\mathrm{arap},i} = \underset{\mathbf{R} \in \mathrm{SO}(3)}{\arg\min} \sum_{j \in \mathcal{N}(i)} \boldsymbol{w}_{ij}^{\mathrm{spatial}} \left\| \mathbf{p}_j' - \mathbf{R} \, \mathbf{p}_j \right\|^2, \tag{6}$$

where $\mathbf{p}_j$ and $\mathbf{p}_j'$ are the relative displacements of neighbor $j$ (w.r.t. Gaussian $i$) in source and target positions, $\boldsymbol{w}_{ij}^{\mathrm{spatial}}$ is a spatial-distance weight ((Eq. 10)), and $\mathcal{N}(i)$ is the $K$-NN set of Gaussian $i$. The rotation $\mathbf{R}_{\mathrm{arap},i}$ is computed via SVD in closed form, with sign correction if $\det(\mathbf{R}_{\mathrm{arap},i}) \leq 0$ to enforce $\mathbf{R}_{\mathrm{arap},i} \in \mathrm{SO}(3)$. The rotation consistency loss is then defined as:

$$\mathcal{L}_{\mathrm{rot}}^{\mathrm{O}} = \mathbb{E}_{t \sim \mathcal{U}[0,1]} \left[ \frac{1}{N} \sum_{i=1}^{N} \left\| \mathbf{R}_{\mathrm{arap},i} \, \mathbf{R}_{0,i} - \mathbf{R}_{t,i} \right\|_F^2 \right], \tag{7}$$

where $\| \cdot \|_F$ denotes the Frobenius norm, and the expectation is approximated by uniform sampling over multiple time instances $t$. This loss enforces consistency between the field-predicted explicit

rotation $\mathbf{R}_t$ and the geometrically derived rigid rotation $\mathbf{R}_{\text{arap}}\mathbf{R}_0$, thereby suppressing non-physical local shearing or twisting and enhancing structural fidelity of the dynamic deformation.

**Hand Loss.** Hand movement in HOI scenes is fast, making dynamic Gaussian fitting much slower and more challenging. Since MANO vertices explicitly represent the position of each point, we design a hand loss to optimize the translation of hand Gaussians. To track their translation, we use a single Chamfer Distance (CHD) to supervise Gaussian translation in 3D space, we compute its distance to the nearest vertex on the MANO $\mathcal{V}_h$. This loss measures the distance from each hand Gaussian to its closest point on the MANO vertices, encouraging the Gaussians to populate the hand surface, formulated as follows:

$$\mathcal{L}_{\text{trans}}^{\text{H}} = \frac{1}{N} \sum_{i=1}^{N} \min_{\mathbf{v} \in \mathcal{V}_h} \left\| (\boldsymbol{x}_i \boldsymbol{y}_i \boldsymbol{z}_i) - (\boldsymbol{x}_v \boldsymbol{y}_v \boldsymbol{z}_v) \right\|_2^2, \tag{8}$$

where $(\boldsymbol{x}_i \boldsymbol{y}_i \boldsymbol{z}_i)$ denotes the $i$-th Gaussian position, and $(\boldsymbol{x}_v \boldsymbol{y}_v \boldsymbol{z}_v)$ represents the filtered points within MANO vertices' range (addressing arm-hand discrepancies).

**Interaction Loss.** Reconstruction of grasping interactions often suffers from edge blurring and mutual occlusion of Gaussians. To regularize the physical reality, we introduce the self-supervised Chamfer distance between hand and object Gaussians. Our approach models the hand and object separately, explicitly defining their positions. This allows us to introduce an interaction loss to ensure proper grasping, formulated as follows:

$$\mathcal{L}_{\text{interaction}} = \frac{1}{\max(|C_H|, \epsilon)} \sum_{i \in C_H} \min_{j \in C_O} \|\mathbf{p}_i - \mathbf{p}_j\|_2^2 + \frac{1}{\max(|C_O|, \epsilon)} \sum_{j \in C_O} \min_{i \in C_H} \|\mathbf{p}_i - \mathbf{p}_j\|_2^2, \quad (9)$$

where $\epsilon = 10^{-6}$ avoids division by zero when no contacts are detected. While this loss promotes hand–object proximity, it does not prevent interpenetration. We therefore use a separate penetration loss (supplementary material) that penalizes overlapping or overly close Gaussians from the hand and object. This loss ensures the physical reality of the interaction and enhances the visual quality by reducing the distance between the hand Gaussians and object Gaussians while preventing overlap.

### 3.4 PROGRESSIVE OPTIMIZATION

In the Hand-Object Interaction (HOI) scene, complex rotations, translations, and occlusions are common. Directly optimizing all Gaussians leads to slow convergence and positional misalignment. To address these issues, we propose a progressive optimization strategy for learning individual implicit fields, which operates in five phases as below: initialization, warm-up, HOI refinement, background optimization, and collaborative reconstruction.

**Initialization.** The MANO vertices Romero et al. (2022) offer a strong prior for hand shape, while the 3D bounding box of object is an effective model-agnostic prior for hinting both hand's and object's initial shapes. To leverage these cues, we initialize $\mathcal{G}_{\text{H}}$ by the derived MANO vertices and initialize $\mathcal{G}_{\text{O}}$ by uniformly sampling within the object's 3D bounding box. For background, we initialize background Gaussians $\mathcal{G}_{\text{BG}}$ from SfM-based points.

**Warm-up.** During the warm-up phase, we use the proposed 3D losses besides the fundamental 2D losses. For the hand field, we employ $\mathcal{L}_{\text{trans}}^{\text{H}}$ to guide the deformation of hand Gaussians $\mathcal{G}_{\text{H}}$, ensuring alignment with the target pose. For the object field, to stabilize interaction-aware transformations, we use $\mathcal{L}_{\text{rot}}^{\text{O}}$. During the warm-up phase, we periodically apply gradient-based density adjustments Kerbl et al. (2023) to optimize the initial Gaussian distribution.

**HOI Refinement.** We adaptively refine Gaussians by assigning each Gaussian $i$ a learnable weight $\boldsymbol{w}_i$ and radius $\boldsymbol{o}_i$, where $\boldsymbol{o}_i$ controls its local influence range. The final refinement weight $\hat{\boldsymbol{w}}_i$ is obtained by: (1) computing spatial proximity weights $\boldsymbol{w}_{ik}^{\text{spatial}}$ for the $K$ nearest neighbors via a Gaussian RBF kernel on distance $\boldsymbol{d}_{ik}$ and $\boldsymbol{o}_i$ (Eq. 10), (2) normalizing these weights to sum to one (Eq. 11), and (3) modulating them with a global importance weight $\sigma(\boldsymbol{w}_i)$ (Eq. 12). This allows joint learning of global importance $\boldsymbol{w}_i$ and local context $\hat{\boldsymbol{w}}_{ik}^{\text{spatial}}$.

$$\boldsymbol{w}_{ik}^{\text{spatial}} = \exp\left(-\frac{\boldsymbol{d}_{ik}^2}{2\boldsymbol{o}_i^2}\right), \quad k \in \mathcal{N}_K(i). \tag{10}$$

Let $\mathcal{N}_K(i)$ denote the set of $K$ nearest neighbor Gaussians for the $i$-th Gaussian, where $\boldsymbol{d}_{ik}$ is the Euclidean distance between the centers of Gaussians $i$ and $k$, and $\boldsymbol{o}_i$ is the learnable radius parameter associated with Gaussian $i$.

$$\hat{\boldsymbol{w}}_{ik}^{\text{spatial}} = \frac{\boldsymbol{w}_{ik}^{\text{spatial}}}{\sum_{j \in \mathcal{N}_K(i)} \boldsymbol{w}_{ij}^{\text{spatial}}} \cdot \tag{11}$$

Finally, the overall refinement weight for Gaussian $i$ is obtained by gating the normalized spatial weight with a learned per-Gaussian parameter $\boldsymbol{w}_i$:

$$\hat{\boldsymbol{w}}_i = \sigma(\boldsymbol{w}_i) \cdot \hat{\boldsymbol{w}}_{ik}^{\text{spatial}}, \quad \text{for} \quad k \in \mathcal{N}_K(i), \tag{12}$$

where $\sigma(\cdot)$ is the sigmoid function ensuring $\boldsymbol{w}_i \in (0, 1)$. This formulation allows the model to learn a global importance weight $\boldsymbol{w}_i$ for each Gaussian while adapting locally based on spatial proximity via $\hat{\boldsymbol{w}}_{ik}^{\text{spatial}}$. Additionally, we query the hand implicit field $\mathcal{F}_{\text{H}}$ and the object implicit field $\mathcal{F}_{\text{O}}$ to obtain their respective rotation matrices $\left(\Delta \boldsymbol{R}_{6D} \in \mathbb{R}^6\right) \rightarrow \left(\Delta \boldsymbol{R} \in \mathbb{R}^{3 \times 3}\right)$ (see Appendix.) and translation offset $\Delta(\boldsymbol{x}_k^t \boldsymbol{y}_k^t \boldsymbol{z}_k^t)$. Using LBS Sumner et al. (2007), we refine the pose of the hand-object Gaussians as follows:

$$\begin{aligned}
\boldsymbol{T}_k^t &= (\boldsymbol{x}_k^t \boldsymbol{y}_k^t \boldsymbol{z}_k^t) + \Delta(\boldsymbol{x}_k^t \boldsymbol{y}_k^t \boldsymbol{z}_k^t), \\
\boldsymbol{\rho}_i^t &= \sum_{k=1}^{3} \hat{\boldsymbol{w}}_i^k \left( \Delta \boldsymbol{R}_k^t ((\boldsymbol{x}_i \boldsymbol{y}_i \boldsymbol{z}_i) - (\boldsymbol{x}_k^t \boldsymbol{y}_k^t \boldsymbol{z}_k^t)) + \boldsymbol{T}_k^t \right).
\end{aligned} \tag{13}$$

Here, $k$ denotes the $k$-th nearest Gaussian to the canonical Gaussian $(\boldsymbol{x}_i \boldsymbol{y}_i \boldsymbol{z}_i)$, and $\boldsymbol{\rho}_i^t$ represents the position of the final Gaussian at timestamp $t$.

**Background Optimization.** We pretrain $\mathcal{G}_{\text{BG}}$ for a fixed number of iterations, performing periodic density control Kerbl et al. (2023) on the background Gaussians to ensure a clean initialization.

**Collaborative Reconstruction.** In the final stage, $\mathcal{F}_{\text{H}}$, $\mathcal{F}_{\text{O}}$, and $\mathcal{F}_{\text{BG}}$ independently deform their Gaussians into a shared target space, enabling full HOI scene reconstruction at any timestamp $t$. Both hand and object Gaussians employ HOI refinement (Eq. equation 13) to update their parameters. The optimization is supervised by interaction constraints $\mathcal{L}_{\text{interaction}}$ (Eq. equation 9) and 2D regularization terms. This stage ensures physically plausible occlusion relationships, smooth edge transitions, and lighting coherence, improving both the geometric fidelity of reconstructed shapes and the temporal smoothness of their motion dynamics.

## 4 EXPERIMENTS

To validate our approach, we conduct comprehensive comparisons with state-of-the-art baselines Wu et al. (2024); Yang et al. (2024); Huang et al. (2024) for dynamic scene reconstruction on both HOI4D Liu et al. (2022) and HO3D Hampali et al. (2020) datasets. Additionally, we compare with HOLD Fan et al. (2024), a specialized method for hand-object interaction reconstruction, on the HO3D dataset. Following Zhang et al. (2024), we evaluate pure translation and translation-rotation using alternate-frame testing to assess extrapolation to novel interactions. Metrics include PSNR, SSIM Wang et al. (2004), and LPIPS Zhang et al. (2018). We further perform full-frame evaluation for completeness (Table 4, appendix). All experiments run on an NVIDIA RTX 3090, achieving optimal performance in 21,000 iterations (1h20m training time).

**Implementation Details.** We employ $K = 3$ nearest neighbors for refinement and deformation, with the key-frame $k$ set to the timestamp just before hand–object contact. Both Gaussians and the deformation model are optimized using Adam. Hyperparameters, schedules, and auxiliary losses (ARAP, elastic, penetration, momentum, 2D) are provided in the supplementary material. HOI4D Liu et al. (2022) provides RGB-D videos with frame-level hand–object poses and masks; we evaluate on two purely translational and two translation–rotation scenes. HO3D Hampali et al. (2020) offers real-world 3D pose annotations for actions like pickup and rotation. We use camera 4 from HO3D and select four translation–rotation sequences for egocentric reconstruction. Both datasets include 3D bounding boxes and MANO hand models.

Tab. 1. Comparison on HOI4D dataset. Best and second best are **bolded** and *italicized* respectively.

| Method | Translation | | | Translation&Rotation | | |
|---|---|---|---|---|---|---|
| | PSNR↑ | SSIM↑ | LPIPS↓ | PSNR↑ | SSIM↑ | LPIPS↓ |
| 4DGS | 24.86 | 0.80 | 0.47 | *23.68* | 0.85 | 0.39 |
| Deform3DGS | *26.33* | *0.87* | **0.29** | 23.57 | **0.89** | **0.25** |
| SC-GS | 25.08 | 0.84 | *0.46* | 17.32 | 0.71 | 0.48 |
| Ours | **30.32** | **0.93** | **0.29** | **24.16** | *0.86* | *0.37* |

Tab. 2. Comparison on HO3D dataset. Best and second best are **bolded** and *italicized*.

| Method | PSNR↑ | SSIM↑ | LPIPS↓ |
|---|---|---|---|
| 4DGS | 19.44 | 0.82 | *0.25* |
| Deform3DGS | 9.68 | 0.36 | 0.65 |
| SC-GS | *20.37* | 0.80 | 0.26 |
| HOLD | 18.03 | *0.84* | 0.26 |
| Ours | **25.19** | **0.89** | **0.15** |

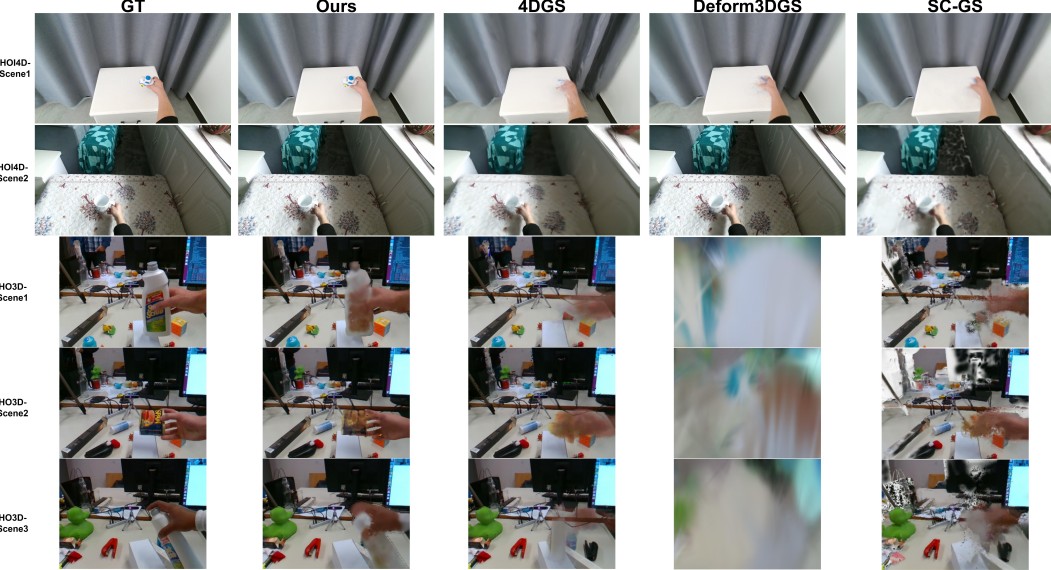

Fig. 3. **Qualitative comparison of our approach and the baseline methods.** We present reconstructions from our model and SOTA baselines (4DGS Wu et al. (2024), Deform3DGS Yang et al. (2024), SC-GS Huang et al. (2024)) on HOI4D and HO3D datasets.

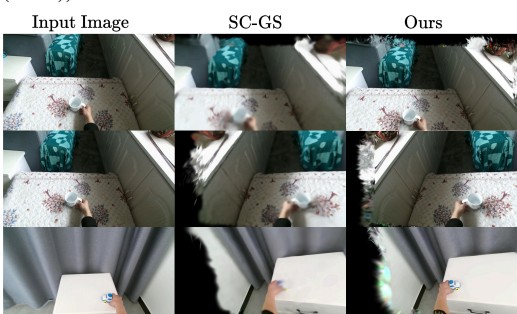

Fig. 4. **Novel view synthesis of our approach and SC-GS.** Our method produces cleaner renderings from novel viewpoints, whereas SC-GS outputs suffer from noticeable noise.

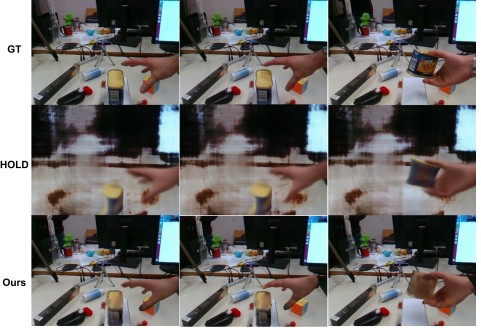

Fig. 5. **Qualitative comparison of our method and HOLD Fan et al. (2024).** Our method achieves complete HOI reconstructions.

## 4.1 QUANTITATIVE COMPARISONS

**HOI4D Dataset.** We compare against 4DGS Wu et al. (2024), Deform3DGS Yang et al. (2024), and SC-GS Huang et al. (2024) using official code and original HOI4D resolution (Table 1). 4DGS is sensitive to initialization and underperforms in HOI settings. Deform3DGS and SC-GS, relying on a single deformation field, fail under occlusion and fast motion. Our interaction-aware, progressively optimized model overcomes these issues, reducing occlusion artifacts and blur while preserving hand-object geometry. We achieve a +9% PSNR gain in translation scenes and improve rotation-heavy scene PSNR from 23.57 dB (Deform3DGS) to 24.16 dB.

**HO3D Datasets.** We downsample all frames to half resolution for efficient processing of large-scale sequences. Despite provided camera and pose estimates, residual calibration errors degrade all methods' metrics. 4DGS is highly sensitive to input noise; SC-GS's sparse control points fail to model background–foreground interactions; and Deform3DGS suffers most due to HO3D's pose

errors (Appendix D of Yang et al. (2024)), causing non-convergence. HOLD Fan et al. (2024), designed for geometry rather than view synthesis, lags behind 3DGS-based methods. Our approach outperforms all baselines via interaction-aware Gaussians and dynamic 3D regularization.

## 4.2 QUALITATIVE COMPARISONS

As shown in Fig. 3 and Fig. 5, our approach surpasses 4DGS Wu et al. (2024), Deform3DGS Yang et al. (2024), SC-GS Huang et al. (2024) and HOLD Fan et al. (2024) in both appearance and shape. In HOI4D-Scene 2, baselines fail to handle Gaussian offsets under dynamic lighting, while our interaction-aware representation preserves shadows and reflections. Isolated background Gaussians improve contrast and dark details. For HOI4D-Scene 1, separate hand-object modeling and 3D losses constrain deformations, with collaborative reconstruction smoothing motion and occlusion. In HO3D-Scene 1 and 2—featuring irregular flipping, rotation, and finger flexibility—4DGS falters under noisy or sparse data and complex motion. SC-GS loses fine details in interaction zones due to sparse control points and handles occlusions poorly. Deformable3DGS, sensitive to pose errors (Appendix D of Yang et al. (2024)), fails to converge on HO3D as errors amplify. HOLD reconstructs hand and object geometry but produces low-quality full-scene renderings. Our method uses $w$ and $o$ to reduce occlusion and blur, achieving superior rendering quality on HO3D.

## 4.3 ABLATION STUDY

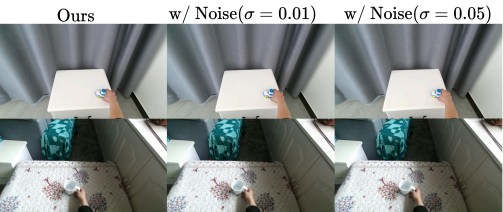

Ours    w/ Noise($\sigma = 0.01$)    w/ Noise($\sigma = 0.05$)

Fig. 6. Our model maintains consistently high rendering quality across different noise levels, with performance nearly matching that of the noise-free setting, showing strong robustness to initialization errors.

Tab. 3. Ablation studies on HOI4D.

| Methods | PSNR↑ | SSIM↑ | LPIPS↓ |
|---|---|---|---|
| w/o HOI Refinement | 32.23 | 0.94 | 0.39 |
| w/o Object Loss | 31.45 | 0.94 | 0.38 |
| w/o Hand Loss | 32.45 | 0.95 | 0.37 |
| w/o Interaction Loss | 31.79 | 0.94 | 0.40 |
| w/o Interaction-Aware | 28.76 | 0.91 | 0.40 |
| w/ noise $\sigma = 0.01$ | 32.80 | 0.95 | 0.35 |
| w/ noise $\sigma = 0.05$ | 32.72 | 0.95 | 0.35 |
| Full Model | **32.96** | **0.95** | **0.35** |

Table 3 reports ablation studies on HOI4D-Scene 1, evaluating the removal of HOI refinement, 3D losses, and the interaction-aware module. We also evaluate robustness to imperfect initialization (e.g., pose or bounding box errors) by adding Gaussian noise $\mathcal{N}(0, \sigma^2)$ to the object's 3D bounding box vertices before point cloud generation, with $\sigma = 0.01$ and $0.05$ in normalized object space. Table 3 and Fig. 6, our full model achieves near noise-free rendering quality under different noise levels. Removing HOI refinement degrades PSNR/SSIM/LPIPS by 2.2%/1.1%/+11.4%; ablating object, hand, or interaction losses causes drops of (4.6%, 1.1%, +8.6%), (1.5%, —, +5.7%), and (3.5%, 1.1%, +14.3%), respectively. Ablating the interaction-aware module—by removing its field parameters and training scheme—leads to a significant performance drop, validating its role in modeling hand–object dynamics.

## 5 CONCLUSION

In this paper, we propose interaction-aware hand-object Gaussians with novel optimizable parameters, adopting piecewise linear hypothesis for a clearer structural representation. This approach effectively captures complex hand-object interactions, including mutual occlusion and edge blur. Leveraging the complementarity and tight coupling of hand and object shapes, we integrate hand information into the object deformation field, constructing interaction-aware dynamic fields for flexible motion modeling. To improve optimization, we propose a progressive strategy that separately handles dynamic regions and static backgrounds. Additionally, explicit 3D regularizations enhance motion smoothness, physical plausibility, and lighting coherence. Experiments show that our approach outperforms the baseline methods, achieving state-of-the-art results in reconstructing dynamic hand-object interactions.

**Limitations.** As designed for interaction modeling, the workflow consists of progressive optimization stages, which could be unified upon emergence of new stronger optimizer. Our method struggles with extreme cases (exceedingly rapid motion/complex trajectories, see Supplement), potentially addressable by integrating more interaction priors.

## ETHICS STATEMENT

This work focuses on 3D reconstruction of human-object interactions (HOI) from publicly available datasets. The research is conducted for academic purposes and aims to advance fundamental understanding in 3D scene understanding and representation learning.

- **Human Subjects and Privacy**: Our method does not involve the collection of new human data. All experiments use existing public datasets (e.g., HOI4D Liu et al. (2022) and HO3D Hampali et al. (2020)), which contain 3D scans or motion-captured sequences of human bodies interacting with objects. Critically, these datasets either exclude facial geometry or have been anonymized (e.g., faces are removed, smoothed, or represented as generic meshes). No personally identifiable information (PII), biometric data, or sensitive attributes are used or reconstructed in this work.

- **Potential Harm**: The proposed approach is a general-purpose 3D reconstruction technique and is not intended for deployment in high-stakes applications such as surveillance, behavioral profiling, or autonomous decision-making. While any 3D human modeling technology could theoretically be misused, our method operates under controlled, indoor settings with coarse body representations and does not recover identity-revealing features.

- **Environmental Impact**: Training and inference were conducted on standard GPU hardware with energy consumption comparable to typical 3D deep learning pipelines. We have made efforts to limit redundant computation through efficient implementation.

- **Dual Use**: We recognize the dual-use nature of human-centric 3D reconstruction. However, since our method does not reconstruct faces, textures, or fine-grained identity cues, the risk of privacy violation or malicious re-identification is minimal.

- **Compliance**: This work complies with the ICLR Code of Ethics. No data was collected unethically, and all third-party datasets are used in accordance with their original licenses and intended research purposes.

We welcome feedback from reviewers regarding any ethical considerations that may have been overlooked.

## REPRODUCIBILITY STATEMENT

We aim to support reproducibility through transparent reporting and planned code release:

- **Code Release**: We intend to release our full codebase—including data loading, training, rendering, and evaluation scripts—along with trained model checkpoints on GitHub. The release will follow a short period for code cleanup and documentation. While it may take a few weeks after the review process, we expect the repository to be publicly available in a timely manner.

- **Datasets**: Experiments are conducted on HOI4D Liu et al. (2022) and HO3D Hampali et al. (2020), using the official data splits and preprocessing protocols. Instructions for data setup will be included in the code repository.

- **Evaluation Metrics**: We report standard image-quality metrics—PSNR, SSIM, and LPIPS—on rendered outputs where ground-truth images are available (e.g., from the same egocentric viewpoint under different conditions or time steps). As is common in egocentric reconstruction, quantitative metrics serve as a proxy for fidelity, while qualitative assessment remains essential.

- **Visual Results**: The main paper includes rendered images of novel view synthesis (NVS) to demonstrate the quality and consistency of our 3D reconstructions. Additional visualizations are provided in the supplementary material.

- **Training**: Trained on a single RTX 3090 (24GB) for 80k iterations ( 3 hours); best results typically appear around 1.5 hours. Implementation uses PyTorch 2.3 with CUDA 11.8. Random seeds are fixed; minor GPU non-determinism may remain.

We believe these details, together with our planned code release, will enable independent reproduction of our work.

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

## A  APPENDIX

The deformation fields $\mathcal{F}_H$ and $\mathcal{F}_O$ predict the following per-Gaussian parameters for their respective sets:

- $\Delta(\mathbf{x}_i \mathbf{y}_i \mathbf{z}_i) \in \mathbb{R}^3$: Translation offset.
- $\Delta \mathbf{R}_i \in \mathbb{R}^6$: 6D rotation representation, which is converted to a $3 \times 3$ rotation matrix $\mathbf{R}_i$ using the method described in supply material.
- $\Delta \mathbf{s}_i \in \mathbb{R}^3$: Scale offset (applied multiplicatively or additively to the base scale).
- $\mathbf{w}_i \in \mathbb{R}$: Learnable weight parameter (as used in Eq. 12).
- $\mathbf{o}_i \in \mathbb{R}^+$: Learnable radius parameter (as used in Eq. 10).

The learnable radius parameter $\mathbf{o}_i$ is distinct from the Gaussian's scale parameters $\mathbf{s}_i$. While $\mathbf{s}_i$ controls the overall size and anisotropy of the Gaussian splat for rendering, $\mathbf{o}_i$ is used solely within the refinement process (Eq. 10) to determine the spatial extent of local neighborhood interactions. The learnable weight $\mathbf{w}_i$ modulates the overall influence of a Gaussian during refinement (Eq. 12) and potentially during rendering or density control, but does not directly alter its geometric scale $\mathbf{s}_i$.

The deformation model is trained using the Adam optimizer with per-parameter learning rates adapted to the geometric scale of the scene. Specifically, for the node-based deformation representation, all trainable parameters (including node positions, radii, weights, and optional rotation parameters) are grouped and assigned an initial learning rate of $\eta_0 = \alpha \cdot \beta \cdot \gamma$, where $\alpha = 0.00016$ is the base position learning rate, $\beta = 5$ is the spatial scale factor (empirically set to the approximate scene extent), and $\gamma = 1$ is the deformation-specific scaling factor. where in practice we use the exponential decay function implemented as:

$$\eta(t) = \begin{cases} \eta_0 \cdot \texttt{lr\_delay\_mult}, & t < t_{\text{warm}} \\ \eta_{\text{final}} + (\eta_0 - \eta_{\text{final}}) \cdot \left(1 - \frac{t - t_{\text{warm}}}{T - t_{\text{warm}}}\right)^p, & t \geq t_{\text{warm}} \end{cases} \tag{14}$$

with $\eta_{\text{final}} = 0.0000016$, $\texttt{lr\_delay\_mult} = 0.01$, and $T = 40{,}000$ maximum steps for deformation parameters. This schedule ensures stable convergence while allowing sufficient exploration during early iterations.

**Full-frame Evaluation.** To evaluate our method more completely, we perform full-frame evaluation (vs. alternate-frame testing in Tab. 4) and introduce two additional metrics: MS-SSIM for structural similarity and ALEX-LPIPS for learned perceptual similarity.

| Metric | PSNR ↑ | SSIM ↑ | MS-SSIM ↑ | LPIPS ↓ | ALEX-LPIPS ↓ |
|--------|--------|--------|-----------|---------|--------------|
| transl | 33.03 | 0.95 | 0.95 | 0.27 | 0.17 |
| r&t | 24.02 | 0.85 | 0.87 | 0.39 | 0.32 |

Tab. 4. Full-frame evaluation on HOI4D, considering both translation (transl) and combined rotation & translation (r&t) tasks.

Tab. 4 shows a complete-frame assessment that provides a more thorough validation of our method's consistency across all temporal frames.

**Gaussian noise in the 3D bounding box.** To evaluate the robustness of our method to imperfect initialization—such as errors in object pose or 3D bounding box estimation—we conduct an ablation study where Gaussian noise $\mathcal{N}(0, \sigma^2)$ is added to the object's 3D bounding box vertices before point cloud generation. Specifically, we perturb the box with $\sigma = 0.01$ and $\sigma = 0.05$ (in normalized object space), simulating realistic inaccuracies in initial object localization. As shown in Table 5

Tab. 5. Experiment on HOI4D-Scene 1.

| Methods | PSNR↑ | SSIM↑ | LPIPS↓ |
|---|---|---|---|
| w/ noise $\sigma = 0.01$ | 32.8019 | 0.9478 | 0.3531 |
| w/ noise $\sigma = 0.05$ | 32.7174 | 0.9477 | **0.3474** |
| Full Model | **32.9579** | **0.9490** | 0.3574 |

Tab. 6. Experiment on HOI4D-Scene 2.

| Methods | PSNR↑ | SSIM↑ | LPIPS↓ |
|---|---|---|---|
| w/ noise $\sigma = 0.01$ | 27.6378 | 0.9029 | 0.2203 |
| w/ noise $\sigma = 0.05$ | 27.4764 | 0.8998 | 0.2369 |
| Full Model | **27.6724** | **0.9049** | **0.2181** |

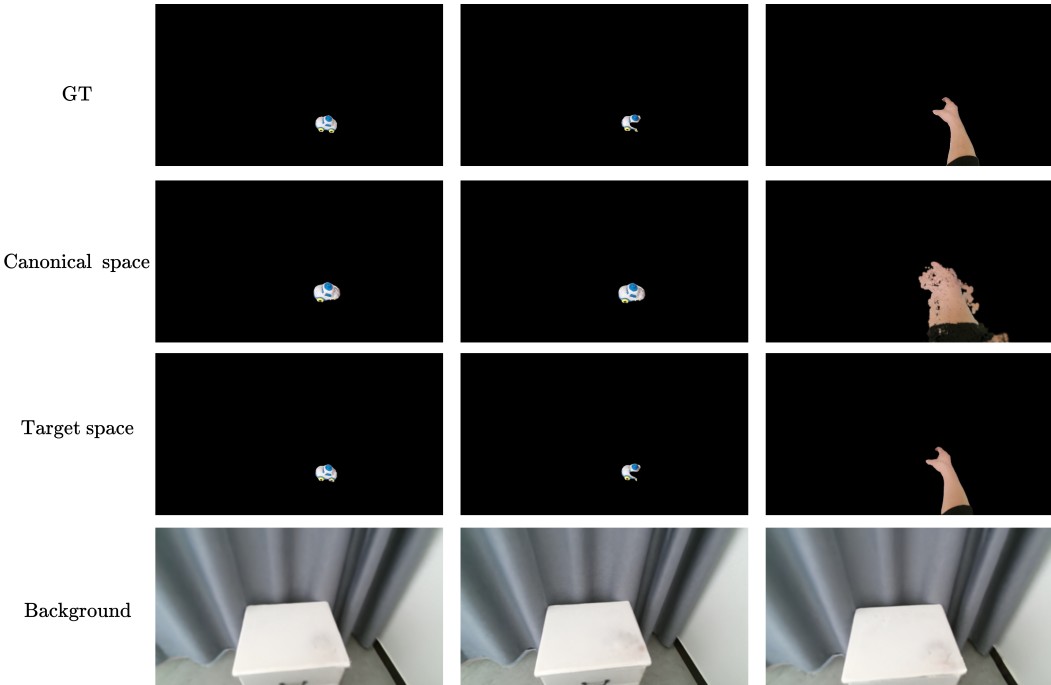

Fig. 7. **Decomposed rendering of object, hand, and background.** We demonstrate the disentangled reconstruction capability of our model by separately rendering the object (left), hand (right), and background (bottom) in both canonical and target spaces. The top row shows ground-truth views for reference. Our method accurately reconstructs each component with consistent geometry and appearance across poses, while preserving scene context in the background.

and Table 6, our method exhibits strong robustness to noisy initial object poses. When Gaussian noise with $\sigma = 0.01$ and $\sigma = 0.05$ is added to the 3D bounding box vertices, the performance degrades only marginally across both scenes. In Scene 1, PSNR drops by merely 0.16 dB (from 32.96 to 32.80) under $\sigma = 0.01$ and by 0.24 dB under $\sigma = 0.05$, while SSIM and LPIPS remain nearly unchanged. Similarly, in Scene 2, the PSNR decreases by only 0.03 dB ($\sigma = 0.01$) and 0.20 dB ($\sigma = 0.05$) relative to the noise-free full model, with consistent trends in SSIM and LPIPS. These

results indicate that our approach is highly insensitive to moderate initialization errors in object localization. We attribute this stability to two key design elements: (1) the interaction-aware optimization that jointly refines hand and object geometry using physical and semantic priors, and (2) the 3D deformation regularizers—specifically the ARAP energy and rotation consistency loss—that constrain the deformation toward physically plausible configurations, effectively suppressing the influence of initialization noise during optimization.

**Separate Presentation of HOI Scenes. Separate Presentation of HOI Scenes.** To demonstrate the disentangled representation learned by our model, we visualize the individual rendering components—object, hand, and background—in both canonical and target views (see Figure 7). Our approach effectively isolates each semantic entity while preserving geometric fidelity and appearance consistency. Notably, the background reconstruction exhibits minimal interference from hand or object motion, suggesting that our model largely succeeds in separating dynamic foreground elements from the static scene context—a desirable property for robust egocentric 3D understanding.

## LLM Usage Statement

We used a large language model (LLM) solely for proofreading and language refinement, such as correcting grammar, improving phrasing, and enhancing clarity of the manuscript. The LLM was not involved in any aspect of research conception, method design, result interpretation, or content generation. All scientific ideas, technical contributions, and experimental analysis presented in this work are entirely the authors' own.

