# OpenReview forum: "Interaction-Aware 4D Gaussian Splatting for Dynamic Hand-Object Interaction Reconstruction"
_ICLR.cc/2026/Conference — ICLR 2026 Conference Withdrawn Submission_

### Official Review · Reviewer_Wic7 · 2025-10-30

**Soundness:** 3
**Presentation:** 3
**Contribution:** 2
**Rating:** 4
**Confidence:** 3

**Summary:**

The paper proposes an interaction-aware 4D Gaussian splatting method for dynamic hand–object interaction reconstruction from RGB video without object shape priors. It decomposes the scene into hand, object, and background fields; conditions the object field on the hand; augments Gaussians with learnable weight (w) and radius (o) to better handle occlusion and sharpen interaction boundaries; and trains with a staged schedule that blends 2D photometric terms with 3D regularizers (e.g., MANO-guided hand constraints, object rotation consistency, and hand–object contact/penetration losses). Experiments on HOI4D and HO3D report state-of-the-art quality over 4DGS-family baselines, with convergence in 21k iterations (1h20m on an RTX 3090).

**Strengths:**

* Practical setup (no category-specific object priors) with a clear hand/object/background decomposition.
* Progressive optimization and interaction-aware constraints appear to stabilize training and improve novel-view quality.

**Weaknesses:**

* Problem statement and method exposition are unclear; the motivation for the new parameters is thin and mostly intuitive.
* Reliance on MANO vertices and an object 3D box undercuts the “no-prior” messaging; availability at test time is unclear.
* Ablations don’t isolate the key interaction design (e.g., hand-conditioned object field) or the specific benefit of the new parameters.

**Questions:**

* Can you provide a controlled study isolating the effect of the interaction parameters (e.g., edge sharpness/occlusion handling)?
* How does the method initialize/converge when 3D cues (MANO/boxes) are unavailable or noisy—can it work from 2D only?
* What is the delta between an object field without hand conditioning and the proposed hand-conditioned version, under identical settings?

---

### Official Review · Reviewer_EF7V · 2025-10-31

**Soundness:** 3
**Presentation:** 3
**Contribution:** 2
**Rating:** 2
**Confidence:** 2

**Summary:**

The paper presents an algorithm for reconstructing 4D Gaussian Splatting from hand-object videos, by modeling the dynamic HOI scenes as hand, object, and background fields separately, along with a series of 2D & 3D losses and a progressive optimization process. The algorithms achieve SOTA rendering performance on two common benchmarks.

**Strengths:**

1. The paper is well-written, easy to understand for readers.
2. The method is technically sound. Also, it achieves SOTA rendering performance on two common benchmarks, compared to existing 4D-GS-based methods.

**Weaknesses:**

1. The difference with previous hand-object reconstruction methods that use independent implicit fields to represent hand/object/background [1, 2] is not clear. Besides replacing SDF/Nerf with Gaussian Splatting, other losses and the optimization strategy are not new things.

2. Metrics on geometry reconstruction accuracy are not reported, like Chamfer Distance, MPJPE/MPVPE, and F-score, as in previous work [1].

3. For reconstruction, comparison with many SOTA works is missed, including G-HOP[2], AlignSDF[3], gSDF[4], iHOI[5], DiffHOI[6], etc.

4. Though they claim that they don’t rely on object prior, the 3D object bounding box is still used for Gaussian points initialization.

[1] HOLD: Category agnostic 3d reconstruction of interacting hands and objects from video, CVPR 2024.

[2] G-HOP: Generative Hand-Object Prior for Interaction Reconstruction and Grasp Synthesis, CVPR 2024

[3] AlignSDF: Pose-aligned signed distance fields for hand-object reconstruction, ECCV 2022.

[4] gSDF: Geometry-driven signed distance functions for 3d hand-object reconstruction, CVPR 2023.

[5] What’s in your hands? 3D Reconstruction of Generic Objects in Hands, CVPR 2022.

[6] Diffusion-Guided Reconstruction of Everyday Hand-Object Interaction Clips, ICCV 2023.

**Questions:**

No.

**Details Of Ethics Concerns:**

No.

---

### Official Review · Reviewer_mDBN · 2025-11-01

**Soundness:** 4
**Presentation:** 3
**Contribution:** 4
**Rating:** 6
**Confidence:** 5

**Summary:**

1. This paper introduces a three Gaussian formulation (hand, object, background) to reconstruct dynamic hand-object interactions.

2. The proposed method employs an interaction-aware strategy that leverages hand cues to refine object Gaussians.

3. The qualitative results show clear improvements over existing approaches.

**Strengths:**

1. The proposed method achieves superior performance in quantitative evaluations compared to existing approaches.

2. A diverse set of prior methods is evaluated against the proposed model.

3. New loss functions are introduced, and comprehensive ablation studies, especially on the Interaction Aware loss, validate their effectiveness.

4. The paper proposes a hand conditioned 3DGS formulation for object modeling.

**Weaknesses:**

1. There is no comparison with BIGS, which also reconstructs meshes from input video.

2. Although the authors claim not to rely on object priors (first contribution: without any object priors), they still use the 3D bounding box of the object:
“In optimization, we utilize explicit 3D information provided by MANO parameters Romero et al. (2022) and the 3D object bounding boxes.”

3. In the novel view synthesis results, the second and third samples appear flipped.

4. The novel view synthesis results seem to involve only translation, without significant rotation.

5. It is unclear why the results on HO3D (which appear noisy and blurry) look worse than those on HOI4D (which appear much more realistic). What accounts for the difference between these two datasets?

**Questions:**

Why is the nearest-neighbor Gaussian count set to 3?

---

### Official Review · Reviewer_Vb1Z · 2025-11-01

**Soundness:** 3
**Presentation:** 2
**Contribution:** 3
**Rating:** 4
**Confidence:** 4

**Summary:**

This paper aims to reconstruct hand-object interaction (HOI) scenes from egocentric videos without relying on any object priors. To this end, it utilizes three 3DGS models to represent the hand, the object, and the background separately. Particularly, the hand Gaussians can be initialized from an off-the-shelf hand mesh estimator, and consequently, the object Gaussians can be inferred based on the hand Gaussians. To better model hand-object interactions, this paper introduces two optimizable per-Gaussian parameters and several explicit 3D-2D regularization losses, along with a multi-stage progressive optimization strategy. The proposed method is validated on several videos selected from HO3D and HOI4D.

**Strengths:**

1. Three separated GS models are used for modeling the hand, the object, and the background, which better handle HOI. And it is reasonable that the background GS model is updated less frequently in the test scenes of this paper.

2. Two learnable Gaussian parameters, including a refinement weight and a radius, are introduced to model interactions.

3. The proposed method outperforms a few baselines on several sequences selected from HO3D and HOI4D.

**Weaknesses:**

1. The comparisons between the baselines and the proposed method are not convincing. Among the four selected baselines, 4DGS, Deform3DGS, and SC-GS are designed for general dynamic scenes, and HOLD adopts the implicit representation and is designed for 3D reconstruction. To demonstrate the superiority of the proposed method, a GS-based method tailored for HOI should be included.

2. Although being better than the selected baselines, the qualitative results of the proposed method still exhibit artifacts like jagged edges and blurred textures (e.g., Fig. 3) and temporal inconsistency (e.g., in the supplemental videos).

3. The selected scenes are of limited diversity and fail to validate the proposed method. For example, the object Gaussians (L224) are introduced to handle deformations like squeezing, but it seems this case is not included in the qualitative results.

4. There are many explicit dynamics 3DGS/4DGS, e.g., ([1, 2, 3]). Particularly, [3] also adopts hand and object Gaussians, and the proposed method differs from it only in the background Gaussians. It would be better if the novelty of the proposed method could be further analyzed.

&emsp;[1] Xuan Huang, et al., Learning interaction-aware 3D Gaussian splatting for one-shot hand avatars. NeurIPS 2024.

&emsp;[2] Gyeongsik Moon, et al., Expressive Whole-Body 3D Gaussian Avatar. ECCV 2024.

&emsp;[3] Chandradeep Pokhariya, et al. MANUS: Markerless Grasp Capture using Articulated 3D Gaussians. CVPR 2025

**Questions:**

1: Figure 1 may be misleading. It is common to predict per-Gaussian deformation for dynamics 3DGS (see Weaknesses 4), which are explicit.

Misc.:

1. Inconsistent symbols, e.g., L076, "$w$, $o$" and "$\bf{w}$, $\bf{o}$".
2. Repeated sentences in L762.

---

### Note · Authors · 2025-11-12

I have read and agree with the venue's withdrawal policy on behalf of myself and my co-authors.